# Dielectric catastrophe at the Wigner-Mott transition in a moiré superlattice

Yanhao Tang[1,2,7] ✉, Jie Gu[1,7], Song Liu[3], Kenji Watanabe[4], Takashi Taniguchi[4], James C. Hone[3], Kin Fai Mak[1,5,6] ✉ & Jie Shan[1,5,6] ✉

The bandwidth-tuned Wigner-Mott transition is an interaction-driven phase transition from a generalized Wigner crystal to a Fermi liquid. Because the transition is generally accompanied by both magnetic and charge-order instabilities, it remains unclear if a continuous Wigner-Mott transition exists. Here, we demonstrate bandwidth-tuned metal-insulator transitions at fixed fractional fillings of a $MoSe_2/WS_2$ moiré superlattice. The bandwidth is controlled by an out-of-plane electric field. The dielectric response is probed optically with the 2s exciton in a remote $WSe_2$ sensor layer. The exciton spectral weight is negligible for the metallic state with a large negative dielectric constant. It continuously vanishes when the transition is approached from the insulating side, corresponding to a diverging dielectric constant or a 'dielectric catastrophe' driven by the critical charge dynamics near the transition. Our results support the scenario of continuous Wigner-Mott transitions in two-dimensional triangular lattices and stimulate future explorations of exotic quantum phases in their vicinities.

Metal-insulator transitions (MITs) accompanied by large electrical conductivity change are widely observed in condensed-matter systems[1–5]. One particularly intriguing origin of charge localization and insulating phases is Coulomb repulsion between electrons[6]. The prototype model for interacting electrons in a lattice is the single-band Hubbard model with electronic bandwidth, $W$, and on-site Coulomb repulsion, $U$. The ground state of the electronic system for half band or full lattice filling is a Mott insulator in the strong interaction limit ($U \gg W$), and a Fermi liquid in the weak interaction limit ($U \ll W$). A MIT, the Mott transition, is expected near $U \sim W$[6]. Similarly, if the electrons are localized by the extended Coulomb repulsion, $V$, instead of $U$, a Wigner-Mott insulator (or a generalized Wigner crystal) that spontaneously breaks the underlying lattice space-group symmetries is formed at fractional lattice fillings[7–10]; a Wigner–Mott transition is expected near $V \sim W$[11–15]. The evolution of a Mott or Wigner–Mott insulator into a metal as a function of the interaction strength remains

a challenging theoretical problem[4,11–15]; it also raises an exciting opportunity for realizing exotic quantum phases near the transition if it is continuous[11,12,14–17].

Experimentally, the interaction strength, or equivalently, the bandwidth can be tuned by applying isostatic or chemical pressure[1]. In almost all known materials, bandwidth-tuned MITs (the case of interest from now on) are driven first order[1] because the MITs are often accompanied by magnetic and structural phase transitions; it is difficult for the various kinds of transitions involving different order parameters to occur simultaneously without fine-tuning[1,14,16]. The recent experimental breakthroughs in semiconducting transition metal dichalcogenide (TMD) moiré materials[7,8,18–20] have opened a new avenue to realize and study continuous bandwidth-tuned MITs[15,21–26]. These materials form a two-dimensional (2D) triangular lattice that suppresses magnetic ordering due to geometric frustration. The electrons are trapped by the periodic moiré potential; they can tunnel

[1]School of Applied and Engineering Physics, Cornell University, Ithaca, NY, USA. [2]Interdisciplinary Center for Quantum Information, Zhejiang Province Key Laboratory of Quantum Technology, and Department of Physics, Zhejiang University, Hangzhou 310027, China. [3]Department of Mechanical Engineering, Columbia University, New York, NY, USA. [4]National Institute for Materials Science, 1-1 Namiki, 305-0044 Tsukuba, Japan. [5]Laboratory of Atomic and Solid State Physics, Cornell University, Ithaca, NY, USA. [6]Kavli Institute at Cornell for Nanoscale Science, Ithaca, NY, USA. [7]These authors contributed equally: Yanhao Tang, Jie Gu. ✉e-mail: yanhaotc@zju.edu.cn; kinfai.mak@cornell.edu; jie.shan@cornell.edu

between the moiré sites and interact with each other via both the on-site and extended Coulomb repulsions. This realizes the triangular-lattice extended Hubbard model[13,21,27,28]. A variety of correlated insulating states are reported, including the Mott insulator[7,18–20] at odd integer filling and Wigner-Mott insulators[7–10,29] at fractional fillings of the moiré superlattice. A continuous bandwidth-tuned Mott transition is also observed by electrical measurements[22,23]. However, the Wigner-Mott transition at fractional fillings remains elusive. In particular, it remains unclear if a continuous Wigner-Mott transition exists given the additional charge-ordering transition (besides possible magnetic transitions) near the MIT[11,14].

Here we report the observation of continuous bandwidth-tuned Wigner-Mott transitions in an angle-aligned MoSe$_2$/WS$_2$ heterobilayer by the exciton sensing technique[8,30]. The heterobilayer forms a triangular moiré lattice with a lattice density of $\approx 1.9 \times 10^{12}$ cm$^{-2}$. It is encapsulated in hexagonal boron nitride (hBN) and gated by a top and bottom few-layer graphite gate (Fig. 1a). The dual-gate structure enables independent tuning of the electron density in the moiré lattice $\nu$ (in units of the moiré density), and the out-of-plane electric field, $E$ (>0 for field pointing from the MoSe$_2$ to WS$_2$ layer). The MoSe$_2$/WS$_2$ heterobilayer has a type-I band alignment with both the conduction and valence band edges located in the MoSe$_2$ layer (Fig. 1b). The band offsets are $\Delta_c \sim 100$ meV for the conduction bands and $\Delta_v \sim 320$ meV for the valence bands from optical spectroscopy measurement (Methods). We tune the electronic bandwidth at a fixed doping density by the electric-field effect[22,23]. The out-of-plane electric field varies the moiré potential depth by controlling the band offset and, correspondingly, the resonance interlayer hopping amplitude[21,22]. Because $\Delta_c < \Delta_v$, the electric-field effect on the conduction bandwidth is much larger; we focus on the case of electron doping.

The dielectric response of the moiré system, which can reflect the critical charge dynamics near continuous MITs[31–33], is probed by the exciton sensing technique. Recent studies show that the technique is highly sensitive to the insulating states[8,30]. These states perturb the electric field between the optically excited electrons and holes (excitons) in a charge-neutral WSe$_2$ monolayer that is separated from the moiré superlattice by a bilayer hBN. The spacer thickness is smaller than the Bohr radius of the 2s and higher-energy exciton states. We

probe the effective dielectric constant, $\varepsilon$, of the moiré heterobilayer by measuring the 2s exciton that has the largest spectral weight. Both the 2s exciton resonance energy and spectral weight, $S$, depend on $\varepsilon$[34]. We analyze the spectral weight near the MITs (Methods). It is difficult to determine the exciton binding energy accurately since the band-to-band transitions are also renormalized and cannot be easily measured. Unless otherwise specified, all measurements are performed at 3.6 K; the corresponding thermal excitation energy is substantially lower than the characteristic energy scales ($U$, $V$ and $W$) of the electronic system. Details on the device fabrication and optical measurements are provided in Methods.

## Results and discussions
### Correlated insulating states in the strong interaction limit

Figure 1c–e shows the reflectance contrast spectrum of device 1 as a function of electron density $\nu$ in the strong interaction limit ($U, V > W$), corresponding to the moiré exciton of MoSe$_2$, the 2s exciton of the sensor layer, and the moiré exciton of WS$_2$, respectively. The moiré exciton spectra are consistent with a previous study[35]; the fundamental exciton in WS$_2$ remains robust for the entire doping range, indicating that the electrons are doped only into the MoSe$_2$ layer in the type-I heterostructure. A series of incompressible states emerge at integer multiples and specific fractions of the moiré density. They modulate the moiré exciton features, but more significantly, the sensor 2s exciton. At each incompressible state, the 2s exciton shows enhanced reflectance contrast or spectral weight (as well as spectral blueshift); it is consistent with small $\varepsilon$. The 2s exciton cannot be identified for the compressible states; it is merged into the band-to-band transitions[30]. The quenching of the 2s exciton is consistent with large negative dielectric constant for a metallic phase.

The insulating states at even integers ($\nu = 2$ and 4) are the single-particle moiré band insulators. The odd integer states ($\nu = 1$ and 3) are the Mott or charge-transfer insulators[27]. The fractional states (e.g. $\nu = 4/3, 3/2, 5/3$ etc.) are the Wigner–Mott insulators. Similar results are reported for a related WSe$_2$/WS$_2$ moiré superlattice[7–9,19]. Generally, with increasing doping density the 2s exciton reflectance contrast decreases, and less fractional states can be identified. It reflects the decreasing importance of the Coulomb interaction at large doping

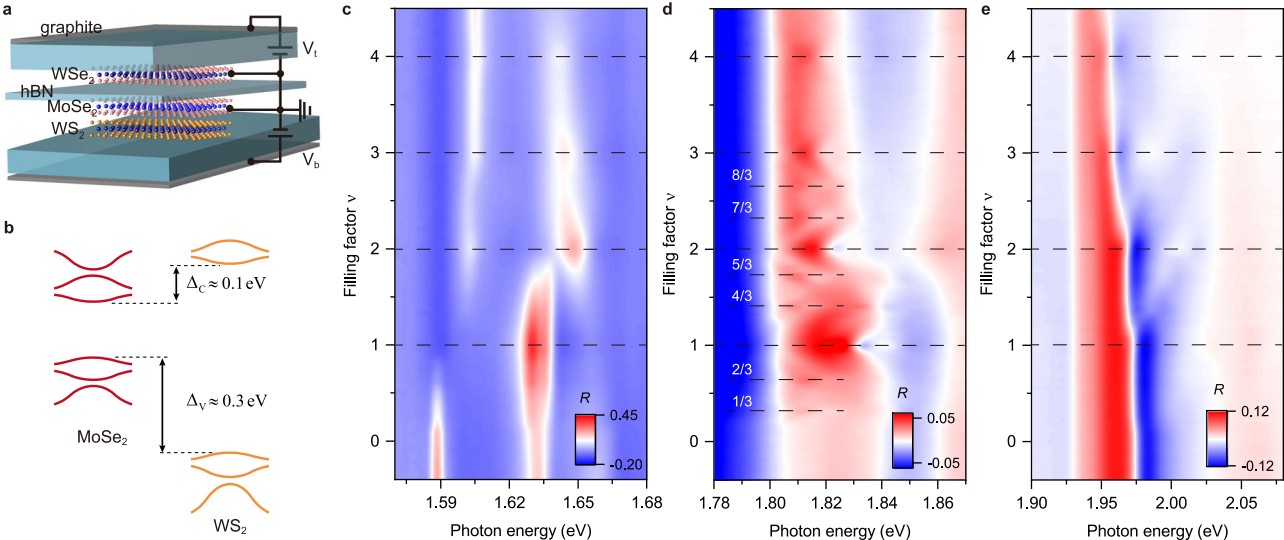

**Fig. 1 | Correlated insulating states in MoSe$_2$/WS$_2$ moiré heterobilayers.**
**a** Schematic illustration of a dual-gated MoSe$_2$/WS$_2$ moiré heterobilayer with an integrated WSe$_2$ monolayer sensor separated by bilayer hBN. Voltage $V_t$ and $V_b$ are applied to the top and bottom hBN-graphite gates, respectively. Both the moiré heterobilayer and the sensor are grounded. **b** MoSe$_2$/WS$_2$ moiré heterobilayer has a type-I band alignment with $\Delta_c \approx 0.1$ eV and $\Delta_v \approx 0.3$ eV. **c–e** Reflectance contrast

spectrum of device 1 as a function of electron doping density v (in units of the moiré density), corresponding to the moiré exciton of MoSe$_2$ (**c**), the sensor 2 s exciton (**d**), and the moiré exciton of WS$_2$ (**e**), respectively. Full lattice fillings are denoted by dashed lines. The incompressible states are identified by the enhanced reflectance contrast of the 2 s exciton.

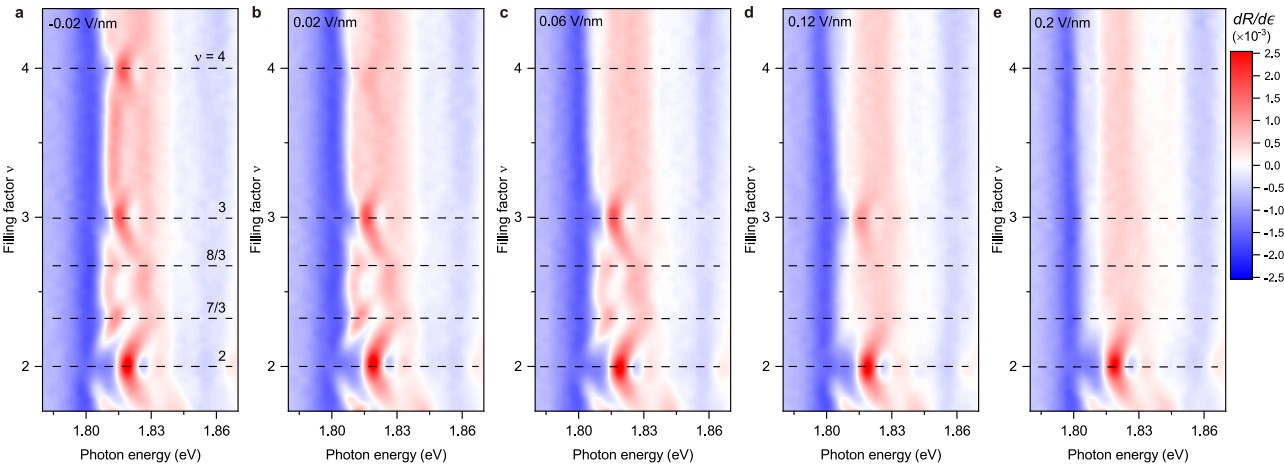

**Fig. 2 | Electric-field-tuned MITs. a–e** The energy-derivative of the reflectance contrast spectrum (**dR/dϵ**) of the sensor 2s exciton as a function of electron filling factor ($\nu = 2$–4) under out-of-plane electric field of −0.02 V/nm (**a**), 0.02 V/nm (**b**), 0.06 V/nm (**c**), 0.12 V/nm (**d**) and 0.2 V/nm (**e**). The incompressible states at $E = -0.02$ V/nm manifest enhanced reflectance contrast and are labeled by the dashed black lines. They gradually disappear as electric field increases.

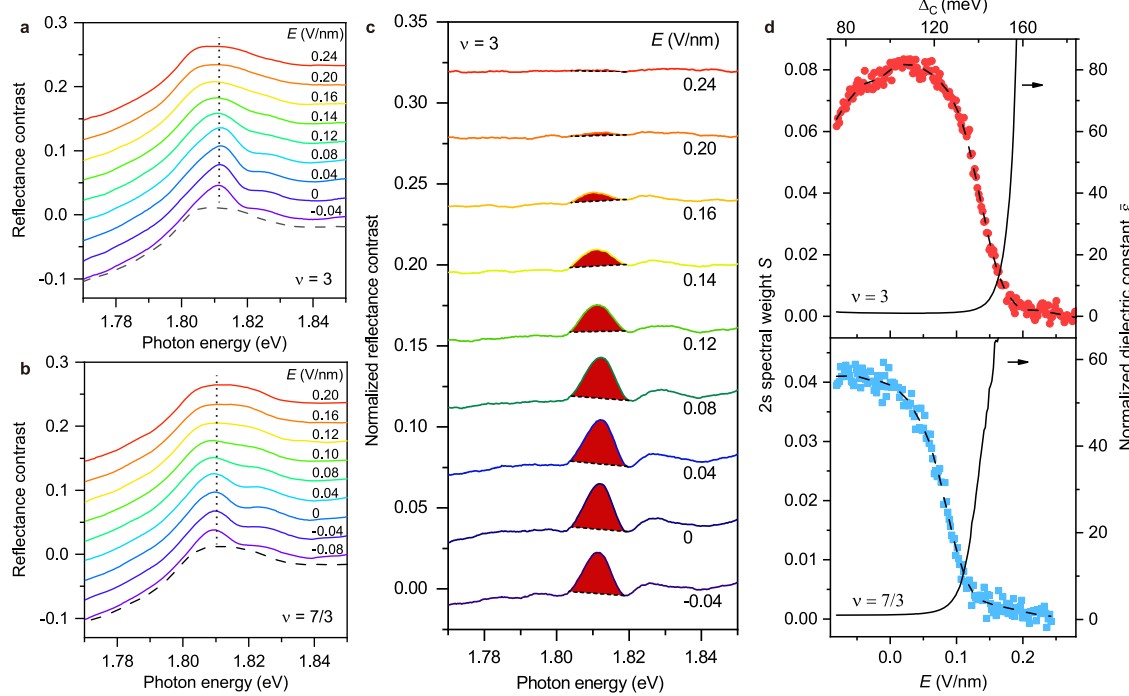

**Fig. 3 | Dielectric catastrophe near the Mott and Wigner–Mott transitions. a**, **b** Reflectance contrast spectrum of the sensor 2s exciton under varying electric fields at fixed electron filling factor of $\nu = 3$ (**a**) and 7/3 (**b**). The spectra are vertically displaced by a constant 0.03. The dashed lines represent the reflectance contrast spectra at high electric fields (0.28 V/nm and 0.24 V/nm for $\nu = 3$ and 7/3, respectively), in which the broad humps correspond to the band-to-band transitions. The vertical dotted lines denote the 2s exciton peak. **c**, Reflectance contrast spectrum at $\nu = 3$, normalized by that at 0.28 V/nm (in the metallic phase), at representative electric fields. The spectral weight $S$ is extracted by integrating the shaded area.

**d** The extracted 2s spectral weight $S$ (symbols) as a function of electric field (bottom axis) and $\triangle_\mathbf{c}$ (top axis) for $\nu = 3$ (upper panel) and $\nu = 7/3$ (lower panel). The band offset $\triangle_\mathbf{c}$ is calculated using the applied electric field as described in Methods. The dashed lines are the smoothed data using the Savitzky-Golay algorithm with a window of 80 mV/nm. The solid lines are the dielectric constant $\bar{\varepsilon}$ of the moiré heterobilayer (right axis) that is normalized to unity at 0.02 V/nm and −0.08 V/nm (upper and lower panel, deep in the insulating phase). It is obtained from the dashed lines using the empirical relation, $\mathbf{S} \propto \varepsilon^{-0.7}$ (Methods).

densities. The second moiré band has a larger bandwidth compared to the first moiré band (Methods). We study the second moiré band with doping density $\nu = 2$–4, for which the field-tuned MITs are more easily achieved. For the same field range, a weak electric-field effect is observed for the first moiré band with $\nu = 0$–2 (Supplementary Fig. 1).

### Bandwidth-tuned metal-insulator transitions
Figure 2 illustrates the evolution of the incompressible states probed by the sensor 2s exciton for $\nu = 2$–4 under increasing electric fields. To enhance the optical contrast of these states, we show the energy derivative of the reflectance contrast spectrum $dR/d\epsilon$. As electric field increases, the insulating states gradually disappear, first the band insulating state at $\nu = 4$, followed by the Wigner–Mott state at $\nu = 7/3$ and 8/3, and the Mott state at $\nu = 3$. The band insulating state at $\nu = 2$ remains robust. The order of disappearance of these states can differ slightly in different devices. The result for device 2 (Supplementary Fig. 2) shows that the fractional states disappear first, followed by the $\nu = 4$ and 3 states; the latter two disappear at similar electric fields.

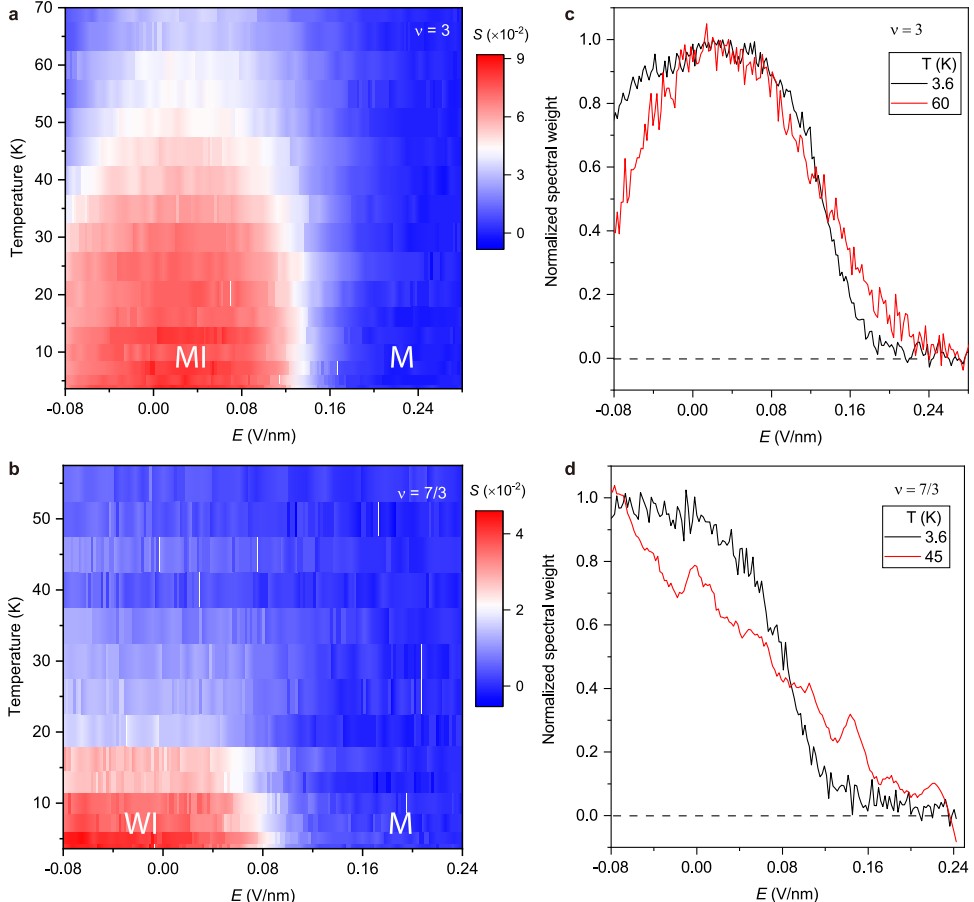

**Fig. 4 | Temperature dependence. a, b** Contour plot of the sensor 2s spectral weight as a function of temperature and electric field for $\nu = 3$ (**a**) and $\nu = 7/3$ (**b**). MI, WI, and M represent, respectively, the Mott insulator, Wigner-Mott insulator, and metal. Regions with enhanced spectral weight (red) correspond to the incompressible states (MI and WI); regions with negligible spectral weight (blue) correspond to the compressible states (metal). **c, d** Electric-field dependence of the 2s spectral weight at selected temperatures for $\nu = 3$ (**c**) and $\nu = 7/3$ (**d**). The spectral weight maximum is normalized to unity. The metal-insulator crossover is broadened at high temperatures.

The above observation is consistent with the bandwidth-tuned MITs. As electric field increases (inducing larger $\Delta_c$ and shallower moiré potential), the moiré bandwidth $W$ increases; this is the predominant effect since $W$ is exponentially dependent on the moiré potential depth[21]. The disappearance of the $\nu = 4$ state indicates closing of the band gap between the second and third moiré band as $W$ increases. The vanishing $\nu = 3$ and the fractional filling states reflect the closing of the Mott charge gap and the Wigner-Mott charge gap when $W$ becomes comparable to $U$ and $V$, respectively. Because $U > V$, the fractional states disappear at smaller critical fields than the $\nu = 3$ states in all devices examined in this study. On the other hand, the relative importance of the Mott gap and the band gap is sample dependent. A plausible origin is the twist angle and moiré density variations since these gaps are generally charge-density dependent due to the strong correlation effects.

Next we investigate the bandwidth-tuned Mott and Wigner-Mott transitions systematically at fixed electron density of $\nu = 3$ and 7/3, respectively. Figure 3a,b illustrate the sensor 2s exciton spectrum as a function of electric field. The 2s exciton resonance vanishes above a critical field, $E_c$. To determine the exciton spectral weight, we first normalize the reflectance contrast spectrum by that at a large field (e.g. 0.28 V/nm, above the critical field for $\nu = 3$). In the metallic phase, the 2s exciton resonance is quenched (Fig. 2); the reflectance contrast spectrum is dominated by a broad hump corresponding to the band-to-band transitions[30]; the spectrum is nearly identical to that at incommensurate fillings. Figure 3c shows

the normalized spectrum at representative fields for $\nu = 3$ (the result for $\nu = 7/3$ is included in Supplementary Fig. 3). The integrated spectral weight (corresponding to the shaded area) is shown in Fig. 3d for $\nu = 3$ and 7/3 as a function of electric field. As electric field or bandwidth increases, the spectral weight continuously decreases to zero. In addition, we do not observe any electric-field hysteresis within the experimental uncertainty. (The $\nu = 3$ spectral weight decreases for negative electric fields because the MoSe$_2$ moiré bands are approaching the WS$_2$ bands.).

We infer the dielectric constant of the moiré heterobilayer from the measured sensor exciton spectral weight by modeling excitons in the 2D sensor layer using realistic device geometry (Supplementary Fig. 4). We numerically solve the electron-hole Schrodinger equation with screened Coulomb potential by the heterobilayer and the hBN substrate with dielectric constant $\varepsilon$ and $\varepsilon_{BN}$, respectively. For $\varepsilon \gg \varepsilon_{BN}$, which holds for the insulting side near the transition, we obtain an empirical relation, $S \propto \varepsilon^{-0.7}$ (Methods). The field-dependence of $\varepsilon$ inferred using the relation is included in Fig. 3d (black lines). Here $\varepsilon$ is normalized by its value deep in the insulating phase, for which the 2s spectral weight plateaus. We also limit the electric-field range such that the signal-to-noise ratio of $S$ stays above 1. We find that $\varepsilon$ increases sharply towards the critical point for both $\nu = 3$ and 7/3; the electric-field dependence of $\varepsilon$ is compatible with a power-law dependence, $\varepsilon \propto |E - E_c|^{-\gamma}$, with exponent $\gamma = 1.6$–2.2 for $\nu = 3$ and 1.2–1.8 for $\nu = 7/3$ (Supplementary Fig. 5). The dependence of $\varepsilon$ on the metallic side cannot be extracted because of the vanished 2s exciton.

The diverging dielectric constant is consistent with the expectations for a continuous Mott/Wigner–Mott transition[32,36]. As the MIT critical point is approached from the insulating side, polarization fluctuations and the holon/doublon density proliferate, giving rise to a 'dielectric catastrophe' $\varepsilon \to \infty$ (Ref. [36]); the charge gap $\Delta \propto \varepsilon^{-1/2}$ vanishes continuously[32]. The result reflects the critical charge dynamics near a continuous MIT and is not expected for a first order MIT. The continuous Mott transition at $\nu = 3$ agrees with recent transport studies of other TMD moiré materials[22,23]. The continuous Wigner–Mott transition at $\nu = 7/3$ is in qualitative agreement with theoretical analyses on the continuous bandwidth-tuned MIT at fractional lattice filling[11,12,14,15]. This transition can proceed through either a two-step process that involves an intermediate charge-density-wave metal[11,14] or a one-step process that involves a deconfined quantum critical point[12,14,15]. Our results are, however, unable to resolve the two scenarios. Additional Hall effect and optical conductivity measurements on the metallic side are required in future studies.

## Crossover at elevated temperatures

Finally, we examine the temperature dependence of these transitions. Figures 4a and 4b show the 2s exciton spectral weight as a function of electric field and temperature at $\nu = 3$ and 7/3, respectively. The spectral weight is always negligible on the metallic side; it gradually decreases on the insulating side with increasing temperature because the thermally excited free carriers in the moiré heterobilayer screen the excitonic interaction in the sensor. The melting temperature is estimated to be ~65 K and 25 K, respectively, for the $\nu = 3$ and 7/3 states. We compare the electric-field dependence of $S$ at 3.6 K and an elevated temperature for the two states in Figs. 4c, d. The spectral weight deep in the insulating phase is normalized to unity. At 3.6 K, the thermal excitation energy is small compared to the charge gap of both the Mott and Mott-Wigner insulators. Compared to the low-temperature behavior at 3.6 K, the MIT becomes a broadened metal-insulator crossover at high temperatures, a manifestation of critical point at lower temperatures. Our result thus suggests either a continuous Mott and Wigner–Mott transition with quantum critical point at zero temperature or weakly first-order transitions with critical point substantially below 3.6 K. The reduced dimensionality, the geometrically frustrated triangular lattice and the presence of disorders are known to favor continuous or weakly first-order transitions[14–16]. Above the critical point, these two scenarios are almost identical[37]; future experiments down to lower temperatures are required to distinguish them.

In conclusion, we have demonstrated bandwidth-tuned Wigner-Mott transitions at fixed band fillings of a Hubbard system based on semiconducting moiré materials. The transitions manifest a dielectric catastrophe when the critical point is approached from the insulating side. Our results present new opportunities to simulate Hubbard physics in the interesting regime of comparable Coulomb repulsion $(U,V)$ and bandwidth $(W)$, and to search for quantum spin liquids near the transitions[12,14–17].

## Methods

### Device fabrication

We fabricate dual-gate devices of a $MoSe_2/WS_2$ moiré heterobilayer with an integrated $WSe_2$ monolayer sensor using the reported dry transfer method[38]. Briefly, atomically thin flakes are first exfoliated from bulk crystals onto Si substrates and then stacked using a polymer stamp to form the desired heterostructure. Monolayer $MoSe_2$ and $WS_2$ flakes are angle aligned with a precision of about 0.5°C. The orientation and relative alignment of these crystals are determined from the angle-resolved optical second harmonic measurement[19]. The $WSe_2$ sensor is separated from the moiré heterobilayer by a bilayer hBN. The TMD moiré heterobilayer and the sensor are grounded through few-layer graphite electrodes. The entire heterostructure is gated by hBN gate dielectrics (≈25 nm) and few-layer graphite gates on both sides.

### Optical reflectance contrast measurements and analysis

Details of the reflectance contrast measurement are reported in the literature[8,19]. Briefly, broadband white light from a tungsten-halogen lamp is focused under normal incidence to a diffraction-limited spot on the device by a high-numerical-aperture objective. The device is mounted in a closed-cycle cryostat with base temperature of 3.6 K (attoDry 1000). The reflected light is collected by the same objective and detected by a spectrometer with a liquid-nitrogen cooled charge-coupled device (CCD). The reflectance contrast spectrum $R \equiv (I' - I)/I$ is obtained by comparing the reflected light spectrum from the sample ($I'$) with a featureless background spectrum ($I$).

To analyze the 2s exciton spectral weight, we integrate the normalized reflectance contrast over a spectral window of 5 nm centered at the resonance. Specifically, we define a straight line that connects the end points of the integration window as a baseline, and integrate the area above it (Fig. 3c). To verify the reliability of the procedure, we have varied the size of the integration window; the process does not affect the electric field dependence of the 2s spectral weight.

### Band alignment of the $MoSe_2/WS_2$ heterobilayer

We determine the band alignment of the $MoSe_2/WS_2$ heterobilayer by examining a sample with large twist angle to avoid the moiré effect for simplicity. The reflectance contrast spectrum is measured as a function of doping density under zero applied electric field. Supplementary Fig. 6a and b show the fundamental exciton resonance in monolayer $MoSe_2$ and $WS_2$, respectively. The neutral exciton feature turns into the charged exciton (or polaron) feature in $MoSe_2$ with both electron and hole doping; the optical response of the $WS_2$ layer remains largely unperturbed. Charges are therefore introduced only into the $MoSe_2$ layer upon both electron and hole doping; the $MoSe_2/WS_2$ heterobilayer has a type-I band alignment.

We determine the band offsets by measuring the electric-field ($E$) dependence of the reflectance contrast spectrum. We choose a fixed electron doping density (≈$3.7 \times 10^{12}$ cm$^{-2}$); the chemical potential is slightly above the conduction band edge of $MoSe_2$. As $E$ increases in the $WS_2$ to $MoSe_2$ direction ($E < 0$), the charged exciton feature in $MoSe_2$ changes to the neutral exciton above $E_0 \approx -0.33$ V/nm (Supplementary Fig. 6c); at the same time, the neutral exciton feature in $WS_2$ turns into the charged exciton (Supplementary Fig. 6d). The spectral changes correspond to the onset of charge transfer from $MoSe_2$ to $WS_2$ when the two conduction bands become nearly degenerate. Using the reported interlayer dipole moment in the $MoSe_2/WS_2$ heterobilayer[35], $d \approx 0.3$ e•nm, we estimate the conduction band offset to be $\triangle_C = d \cdot E_0 \approx 0.1$ eV. The valence band offset can be evaluated as $\triangle_V \approx E_g^W - E_g^{Mo} - \triangle_C \approx 0.32$ eV, where $E_g^W \approx 2.04$ eV and $E_g^{Mo} \approx 1.62$ eV are the optical gaps of monolayer $MoSe_2$ and $WS_2$, respectively. Supplementary Fig. 6e illustrates the inferred band alignment.

### Estimate of the bandwidth

We estimate the first moiré conduction bandwidth, $W_0 \sim \frac{\hbar^2}{ma_M^2}$ ~5–10 meV (corresponding to a temperature scale ~60–120 K), from the moiré period $a_M$. Here $\hbar$ and $m$ denote the Planck's constant and the conduction band mass of monolayer $MoSe_2$, respectively ($m \approx 0.56$ $m_0$ in terms of the free electron mass $m_0$)[39]. The combined width of the first two moiré bands is $4W_0$; the second moiré bandwidth is about $3W_0$ (~180–360 K). These values are substantially smaller than $\triangle_C \approx 0.1$ eV (~1200 K). We focus on the second moiré conduction band with larger bandwidth so that the sample temperature is small compared to the bandwidth and the bandwidth-tuned MITs are easier to achieve.

The electrons reside in the MoSe$_2$ layer for all electric fields and doping densities in this study.

## Modeling the 2s exciton of the sensor layer

Quantitative estimate of the effective dielectric constant of the moiré heterobilayer is obtained by modeling the sensor 2s exciton (Supplementary Fig. 4). We solve the Schrödinger equation, $H\Psi_{ns}(\rho) = E_{ns}\Psi_{ns}(\rho)$, for the energy ($E_{ns}$) and wavefunction ($\Psi_{ns}(\rho)$) of the $n$s ($n$ = 1, 2, ...) exciton state using the finite difference method[40]. For radially symmetric $n$s excitons confined in the 2D sensor plane, the Hamiltonian is given by $H = -\frac{\hbar^2}{2m_R}\left(\partial_\rho^2 + \frac{1}{\rho}\partial_\rho\right) + V(\rho)$, where $\rho$ is the distance between the electron and hole, $m_R$ ($\approx 0.2\,m_0$)[41] is the reduced mass of the exciton, and $V(\rho)$ is the electrostatic potential between the electron and hole. We model $V(\rho)$ using the device geometry shown in Supplementary Fig. 4a. The thickness of the sensor layer, which is substantially smaller than the 2s exciton Bohr radius (~5 nm), is ignored; the thickness of the hBN spacer and the moiré heterobilayer are $d_1$ and $d_2$, respectively; the hBN gate dielectric is assumed to be infinitely thick. The latter is a good approximation when the exciton Bohr radius does not exceed substantially the hBN thickness and screening by the graphite gates is negligible. We express the potential as follows[42]

$$V(\rho) = -\frac{e^2}{4\pi\varepsilon_0\varepsilon_{BN}}\left[-\frac{1}{\rho} + \sum_{j=0}^{\infty}\beta^{2j+1}\left(\frac{1}{\sqrt{\rho^2 + 4(d_1 + jd_2)^2}} - \frac{1}{\sqrt{\rho^2 + 4(d_1 + (j+1)d_2)^2}}\right)\right].$$
(1)

Here $\varepsilon_0$ is the vacuum permittivity; $\beta = (\varepsilon - \varepsilon_{BN})/(\varepsilon + \varepsilon_{BN})$ is given by the dielectric constant of hBN ($\varepsilon_{BN}$ = 4.5)[8] and the heterobilayer ($\varepsilon$); we take $d_1$ = 0.9 nm for a 2L-hBN spacer and $d_2$ = 0.6 nm for electrons residing in the MoSe$_2$ layer of the moiré. We assume $\varepsilon$ to be a real value for the insulating states for simplicity. For $\rho$ larger than the 1 s exciton Bohr radius (<2 nm), the potential $V(\rho)$ in Eq. (1) is a good approximation for the more accurate Rytova-Keldysh potential for 2D excitons[40,43,44] (Supplementary Fig. 7). Since we focus on 2s sensing in this study, Eq. (1) is sufficient; we can ignore the finite sensor thickness.

Supplementary Fig. 4b, c illustrate the spatial distribution of potential $V(\rho)$ and wavefunction $\Psi_{2s}(\rho)$ of the 2s exciton for several values of $\varepsilon/\varepsilon_{BN}$. The 2s exciton radius $r_{2s}$ ($=\sqrt{\langle\Psi_{2s}|\rho^2|\Psi_{2s}\rangle}$) and binding energy $E_{2s}$ as a function of $\varepsilon/\varepsilon_{BN}$ are shown in Supplementary Fig. 4d, e, respectively. For $\varepsilon/\varepsilon_{BN}$ = 2, screening by the heterobilayer is negligible; we have $V(\rho) \approx -\frac{e^2}{4\pi\varepsilon_0\varepsilon_B\rho}$ and $r_{2s} \approx$ 4.7 nm. The latter agrees well with the reported value of $r_{2s} \approx$ 6.6 nm for monolayer WSe$_2$ embedded in hBN[40]. As $\varepsilon/\varepsilon_{BN}$ increases, $V(\rho)$ is suppressed and $\Psi_{2s}(\rho)$ is flattened; $r_{2s}$ increases and $E_{2s}$ decreases. For $\varepsilon/\varepsilon_{BN}$ = $10^4$, $r_{2s}$ exceeds 50 nm. We limit the range of $\varepsilon/\varepsilon_{BN}$ to <$10^4$ (so that the correction from the gate screening effect remains small) and perform a power-law analysis of $E_{2s}$. The binding energy is well described by $\sim(\frac{\varepsilon}{\varepsilon_{BN}})^{-0.7}$ for $\frac{\varepsilon}{\varepsilon_{BN}} > 10$ (solid line in Supplementary Fig. 4e). The exciton spectral weight is expected to follow the same scaling law on the dielectric constant since both the exciton spectral weight and binding energy scale quadratically with inverse of the exciton radius, which represents the total number of electronic states constituting the exciton[45] (Supplementary Fig. 4f shows $E_{2s} \sim (r_{2s})^{-2}$ for the entire range of dielectric constant). We examine the result for several hBN space thicknesses. The power-law exponent of −0.7 remains a good approximation as long as the spacer thickness is much smaller than the 2s exciton radius. In the main text, we use the power-law dependence to extract the evolution of the dielectric constant on the out-of-plane electric field from the experimental spectral weight.

## Data availability

Source data are provided. Additional data that support this work are available upon reasonable request to the corresponding authors. Source data are provided with this paper.

## Code availability

All codes to analyze the reflection contrast spectra are available upon reasonable request to the corresponding authors.

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

## Acknowledgements

We thank Chenhao Jin for fruitful discussions. The research was supported by the National Science Foundation (NSF) under DMR-2114535 (development of the sensing technique) and the US Army Research Office under grant number W911NF-17-1-0605 (device fabrication). Growth of the $MoSe_2$ and $WSe_2$ crystals was supported by the U.S. Department of Energy (DOE), Office of Science, Basic Energy Sciences (BES), under Award # DE-SC0019481 and growth of the hBN crystals by the Elemental Strategy Initiative of MEXT, Japan and CREST (JPMJCR15F3), JST. This work made use of the Cornell NanoScale Facility, an NNCI member supported by NSF Grant NNCI-2025233. K.F.M. acknowledges support from the David and Lucille Packard Fellowship. Y.T. acknowledges support from a startup funding by Zhejiang University.

## Author contributions

Y.T. and J.G. fabricated the devices and performed the optical measurements. S.L. and J.H. grew the bulk TMD crystals. K.W. and T.T. grew the bulk hBN crystals. Y.T., K.F.M., and J.S. designed the study, performed the analysis, and co-wrote the manuscript. All authors discussed the results and commented on the manuscript.

## Competing interests

The authors declare no competing interests.
