## [Peer Review File · Nature Communications]

Dielectric catastrophe at the Wigner-Mott transition in a moiré superlatticeEditorial Note: This manuscript has been previously reviewed at another journal that is not operating a transparent peer review scheme. This document only contains reviewer comments and rebuttal letters for versions considered at *Nature Communications*.

Reviewer #1 (Remarks to the Author):

The authors present a very nice optical study where the excitonic state in a remotely positioned WSe₂ layer interacts with the dielectric environment of a MoSe₂/WS₂ bilayer that is undergoing a metal insulator transition.

This work is a natural extension of the previous publications from this group that a) have reported the MIT in this class of Moire lattices at integer and fractional filling, and b) used the 2s exciton feature as a "sensor" for the phase transition. Previous works (Nature 587, 214(2020)) have identified that the insulating state at fractional fillings is more delicate than that at integer fillings of the Moire band.

Therefore, I interpret the main experimental statement of this paper to be the measurement of the dielectric constant (catastrophe) across the MIT at both integer and fractional fillings.

I feel that alone is probably insufficient to warrant publication in Nature Materials as it is a natural continuation of previous works without providing new fundamental (qualitative) insight. However, in order to better my understanding of the work I'd like to ask the authors to answer the following:

Response 1:

We thank the reviewer for the positive assessment of the quality of our work. We agree that we have not provided enough background in our original manuscript and have not clearly explained the importance of our study. As pointed out by the reviewer, our earlier study has demonstrated the exciton sensing technique and its application on detecting insulating states at fractional filling factors; we have also demonstrated bandwidth-tuned metal-insulator transition (MIT) at integer fillings of the moiré lattice in AA-stacked MoTe₂/WSe₂ moiré bilayer. However, the electrical contacts in our earlier study were not good enough to perform transport studies at fractional fillings less than $\nu = 1$. As far as we know, there is no experimental report of bandwidth-tuned continuous MIT at fractional fillings of the moiré lattice, i.e. the continuous Wigner-Mott transition. Our work here provides the first experimental evidence of such transition.

The existence of a continuous Wigner-Mott transition is not obvious from a theoretical standpoint (we only focus on bandwidth-tuned transitions here). Because MITs are typically accompanied by magnetic, structural and other forms of phase transitions, they are driven first-order without fine-tuning of the different transitions to happen at the same critical point [e.g. PRB 78, 045109 (2008), PRB 91, 235140 (2015), arXiv:2106.14910, arXiv:2111.09894, PRL 113, 197202 (2014)]. For instance, Mott transitions are typically accompanied by

antiferromagnetic phase transitions because Mott insulators are antiferromagnetic in general. As a result, most of the Mott transitions are driven first-order [e.g. Rev. Mod. Phys. 70, 1039–1263 (1998)]. One route to realize a continuous Mott transition (without fine-tuning) is to suppress the antiferromagnetic order. A geometrically frustrated lattice (e.g. a triangular lattice) is therefore a promising place to look for such transitions [e.g. PRB 78, 045109 (2008), PRB 91, 235140 (2015), arXiv:2106.14910, arXiv:2111.09894, Nature 464, 199–208 (2010)].

Compared to the Mott transition, the Wigner-Mott transition is also accompanied by a charge-order phase transition (in addition to possible antiferromagnetic transition). (The Wigner-Mott insulator is a charge-ordered state that breaks the translational symmetry of the lattice while the metallic state does not.) Therefore, even if the antiferromagnetic order is suppressed by geometric frustration, the charge-order phase transition always remains; it is unclear if a continuous Wigner-Mott transition is possible. Theoretical studies [e.g. Nature Physics 4, 932-935 (2008), arXiv:2106.14910, arXiv:2111.09894, PRL 113, 197202 (2014)] suggest that it is possible and there are two routes to achieve such transition: 1) a two-step transition from a charge-ordered insulator to a charge-density wave (CDW) metal and finally to a Fermi liquid without charge-order; and 2) a one-step transition from a spin liquid to a Fermi liquid through a deconfined quantum critical point.

Given this background, it is therefore important to show the possibility of a continuous (bandwidth-tuned) Wigner-Mott transition. Our observation of a dielectric catastrophe at fractional fillings of the moiré lattice (i.e. a continuously vanishing charge gap) suggests that such transition exists. Although we are not able to tell if the transition is a two-step or a one-step process without other complementary measurements (e.g. the Hall effect), we believe that it is an important step towards solving this problem. In our revised introduction (page 1 and 2), we have provided additional background on the problem of Wigner-Mott transition in order to bring our work to the context (changes are highlighted in yellow).

1) I think the authors should add a discussion on why observing only the insulator-side of the dielectric catastrophe is sufficient. Am I correct in understanding that the 2s exciton peak is absent in the metallic phase, and therefore it is not possible to measure the dielectric constant by the same methods? Are there other possible approaches to measuring the permittivity across the MIT?

Response 2:

We thank the reviewer for the question. The reviewer is correct that because the 2s exciton is fully screened by the metallic state, it is not possible to use the exciton sensing technique to obtain the dielectric constant on the metallic side (which is expected to be a large negative number). However, it is still possible to observe a dielectric catastrophe and therefore a continuous Wigner-Mott transition even without knowing the metallic dielectric constant. The reason is that the charge gap closes continuously for such transition, corresponding to a diverging dielectric constant as the critical point is approached from the insulating side. But if one could also measure the dielectric constant of the metallic state, it would be possible to tell

if the transition is a one-step or two-step process as discussed in **Response 1**. A possible way to measure the dielectric function of the metallic state is to measure the complex ac Drude-like conductivity through far-infrared spectroscopy. Such measurement is technically challenging for the mesoscopic devices used in this study. Additional studies are required to measure the metallic dielectric constant.

In our revised manuscript (page 5), we have included a short discussion on why measuring the dielectric constant on the insulating side alone is sufficient to conclude the presence of a continuous Wigner-Mott transition. We have also discussed other possible measurements on the metallic side for future studies.

2) I feel a bit uneasy about the statement: "In almost all known materials, the MIT is driven first order." I can't help but feel this is written in a manner to oversell the results on TMDs. I surely agree that many MIT in nature are associated with lattice distortions, etc, but there are now a multitude of two-dimensional electronic systems that display tunable electronic MITs. The authors here portray the physics in their system to be very elusive, and while I don't disagree that the degree of tunability is exceptional in TMDs, I'm not sure that is the case. I believe a more honest discussion on the state of the field is warranted.

Response 3:

We agree with the reviewer that many 2D electron systems are now known to exhibit continuous MITs. But in almost all cases they are density-tuned MITs rather than bandwidth-tuned MITs [e.g. Rev. Mod. Phys. 82, 1743-1766 (2010), Rev. Mod. Phys. 73, 251-266 (2001)]. We believe that there are only a few systems that show bandwidth-tuned continuous MITs [e.g. https://doi.org/10.36471/JCCM_September_2021_03]; these include organic Mott insulators, TMD moiré materials and possibly ABC trilayer graphene moiré. All of these systems share a triangular or close to a triangular lattice structure, in which the geometric frustration suppresses antiferromagnetic ordering. As discussed in **Response 1**, a continuous bandwidth-tuned MIT at a fixed integer or fractional filling is rare because of the accompanied magnetic and/or structural phase transitions. We agree that our statement was not very accurate. It was intended for bandwidth-tuned MITs. In our revised manuscript (page 2), we have specifically referred to bandwidth-tuned MITs in order to avoid confusion.

3) Three mentions of exotic spin-liquid phases near the transition seems a bit much.

Response 4:

We agree with the reviewer. We have now reduced the emphasis on quantum spin liquids (only once in the conclusion paragraph).

4) The paragraph ending with "However, the Wigner transition remains elusive." is confusing for me as the authors mention the observation of a Mott-Wigner transition in the preceding sentence. What specifically is elusive and has been observed for the first time in this work?

Response 5:

We thank the reviewer for the comment. We did not say in the preceding sentence that the Wigner-Mott transition has been observed. What we have said is that both the Mott insulator and the Wigner-Mott insulator (or the generalized Wigner crystal) have been observed by earlier studies. Only the Mott transition (i.e. the MIT at filling factor $\nu = 1$ of the moiré lattice) has been demonstrated so far; the Wigner-Mott transition (i.e. the MIT at fractional fillings) has not been reported yet. We believe that our study is the first to report a bandwidth-tuned Wigner-Mott transition at a fixed fractional filling. That is why we said it remains elusive. We have improved our introduction paragraphs to make the message clearer.

5) More generally, I'm unclear on the different physical implications of a Mott vs Mott/Wigner ground state emerging at integer/fractional fillings. I see only quantitative differences in the data (obviously the filling factor is different, and also the "melting" temperature). Are there qualitative differences that should be expected? How does a Wigner transition in a Moire potential compare with that of a free electron gas?

Response 6:

We thank the reviewer for the question. We believe that we already gave most of the answers to this question in **Response 1**. To summarize, a Mott transition at $\nu = 1$ preserves the translational symmetry of the moiré lattice; the triangular lattice structure is maintained on both the insulating and metallic sides. The geometrically frustrated lattice helps suppress antiferromagnetic ordering for the Mott insulator and therefore makes a continuous Mott transition possible. In contrast, a Wigner-Mott transition at fractional fillings involves translation symmetry breaking; the Wigner-Mott insulator breaks the translational symmetry of the triangular lattice but the metallic state does not. Even if antiferromagnetic ordering is suppressed by geometric frustration, it is unclear if a continuous Wigner-Mott transition is possible in the presence of charge ordering. Recent theoretical studies argue that such transition is possible [e.g. Nature Physics 4, 932-935 (2008), arXiv:2106.14910, arXiv:2111.09894, PRL 113, 197202 (2014)] and there are two scenarios for this to occur: 1) a two-step transition first from a charge-ordered insulator to a metallic CDW and then from the CDW to a Fermi liquid; and 2) a one-step transition from a spin liquid to a Fermi liquid through a deconfined quantum critical point. Our observation of a dielectric catastrophe for the Wigner-Mott transition, which implies a continuously vanishing charge gap from the insulating side of the transition, shows that such transition can be continuous; we believe that this is an important finding.

The reviewer is correct that the Mott transition and the Wigner-Mott transition, if both are continuous, are only quantitatively different in terms of the exponent for the power-law dependence of the dielectric constant and charge gap near the critical point. However, the

Wigner-Mott transition can go through a two-step process; this is qualitatively different from the Mott transition. It is important to know if the transition is a one-step or two-step process. Unfortunately, our experiment is only able to tell that it is a continuous transition but cannot tell the two scenarios apart. Measurement of the Hall coefficient through the transition in future studies will help better understand the nature of the observed transition. In particular, the Hall coefficient is expected to jump at the transition from a metallic CDW with a reconstructed Fermi surface to a Fermi liquid that breaks no symmetry.

The Wigner-Mott transition is also different from the Wigner transition for an electron fluid in continuum. Such electron fluid is expected to go through a magnetic transition before the transition to a Wigner crystal as the interaction strength increases [e.g. Rev. Mod. Phys. 82, 1743-1766 (2010), Rev. Mod. Phys. 73, 251-266 (2001)]; the interaction strength is typically tuned by the electron density. The magnetic ground state of the Wigner crystal near the Wigner transition is also different; theoretical studies suggest that it could be a quantum spin liquid [e.g. Phil. Mag. B 79, 859, (1999)]. From a symmetry perspective, the Wigner transition in continuum breaks the translational symmetry of the vacuum; the Wigner-Mott transition only breaks the translation symmetry of the lattice. A much larger potential-to-kinetic energy ratio is needed for the former; the presence of a lattice lowers this ratio for the latter.

In order to better bring our work to the context, we have provided in the introduction additional backgrounds on the Mott and Wigner-Mott transitions.

6) On the estimation of bandwidth discussion in the methods section - I'd appreciate if the authors also include temperature as a relative energy scale (melting T of 50 K \sim 5 meV and seems to be similar in magnitude to the band width. I would have assumed $T_{\text{experiment}} \ll T_F$ would be a prerequisite for observing the physics)?

Response 7:

We have now included the temperature scale in the revised Methods (page 7). The reviewer is correct that the experimental temperature (down to 3.5 K) is much smaller than the bandwidth of the second moiré band (about 180-360 K). Meanwhile, the bandwidth of the first moiré band is about 60-120 K. We only focus on the second moiré band in this study because of its larger bandwidth.

Reviewer #2 (Remarks to the Author):

In this work, the authors study the metal to insulator transition in a heterostructure of monolayer MoSe₂ and WS₂ using the exciton sensing technique. They demonstrate that by tuning the band alignment of the substituent layers with a displacement field, the bandwidth of the Moire band can be continuously tuned, thus switching the ground state of the superlattice from a Mott (Wigner) insulator to a metallic state, a phenomenon that has been studied previously by them in a different heterostructure (MoTe₂/WSe₂).

Although the new result and analysis is convincing to me, I find the major claim, the existence of a dielectric catastrophe across the phase transition, does not have significant enough impact to be published in Nature Materials. Since by definition any metal to insulator phase transition involves a large increase in electrical conductivity, it does not surprise me that tuning out of the Wigner insulator state can cause a large change in the dielectric constant, according to the Kramers Kronig relation.

Given that both the continuous phase transition in the TMD heterostructure and the exciton sensing technique have been reported before by the same group, I find the major novelty of this work lies in reporting on another TMD heterostructure exhibiting a similar MIT phenomenon, which is on the incremental end among the great works the authors have published.

Response 8:

We thank the reviewer for the comments that help improve our manuscript. We believe that we have not provided enough background for our work so that its importance has been overlooked. We have provided the detailed scientific background in **Response 1**. To summarize, our work on MoTe₂/WSe₂ has only demonstrated the Mott transition at fixed filling factor $\nu = 1$ of the moiré lattice; the bandwidth-tuned Wigner-Mott transition at fixed fractional fillings has not been demonstrated as far as we know. The Wigner-Mott transition is different from the Mott transition because translation symmetry breaking is involved. Therefore, even if the antiferromagnetic phase transition is fully suppressed by geometric frustration in a triangular lattice, it is not clear if a continuous Wigner-Mott transition is possible. Recent theoretical studies have suggested that such transition is possible [e.g. Nature Physics 4, 932-935 (2008), arXiv:2106.14910, arXiv:2111.09894, PRL 113, 197202 (2014)] but the physics involved is very different from the Mott transition (details please refer to **Response 1**). In this context, our observation of a dielectric catastrophe for the Wigner-Mott transition, which implies a continuously vanishing charge gap from the insulating side of the transition (i.e. a continuous transition), is an important result. For comparison, the transition could well be a first-order abrupt transition. No dielectric catastrophe is expected in that case because there is no critical dynamics; the charge gap would just drop abruptly to zero at the transition. We therefore believe that our work is not merely reporting the same phenomenon in a different material system. To improve our manuscript, we have provided additional background on the problem of Wigner-Mott transition in the introduction paragraphs (page 1 and 2, changes are highlighted in yellow).

To help improve the manuscript, below I include two technical comments for the authors to consider.

1. The approximation that the sensing layer has zero thickness and that the Coulomb potential in it would have the same form as in the free space, if there were no MIT layer, seems too strong to me. It is well known that when the exciton radius is comparable to the 2D semiconductor thickness, the Rytova-Keldysh potential becomes more applicable. I expect this to be the case when the system is in the insulating phase, since the radius of 2s exciton, the thickness of the sensing layer, the spacer layer, and the heterostructure are all on the same level.

Response 9:

We thank the reviewer for the comment. We agree with the reviewer that the Rytova-Keldysh potential is more accurate compared to the Coulomb potential. This is especially the case for 1s excitons, in which the short-distance physics is the most relevant. At large distances (compared to the 1s exciton Bohr radius ~ 2 nm), however, the Coulomb potential becomes a good approximation of the Rytova-Keldysh potential (Fig. R1a). The deviation of the exciton binding energy for the n -th Rydberg state from the correct value becomes smaller and smaller as the principle quantum number n and the n -th exciton Bohr radius increase (Fig. R1b). In practice, the 2s Bohr radius is ~ 5 nm without the MIT layer, which is already substantially larger than the sensor thickness ~ 0.6 nm (the 2s Bohr radius is expected to be larger than 5 nm in the presence of the MIT layer, especially near the MIT). The Coulomb potential becomes a good approximation for $n \geq 2$. This has also been pointed out by earlier studies [e.g. PRL 120, 057405 (2018), PRL, 113, 076802 (2014), PRL 113, 026803 (2014)]. Since we only employed the 2s exciton for sensing, the analysis based on the Coulomb potential is a good approximation. In the revised Methods (page 7), we have justified the use of the Coulomb potential for 2s exciton sensing. We have also included Fig. R1 as Extended Data Fig. 7.

Figure R1. Comparison between the Coulomb and Rytova-Keldysh potential. a, The Coulomb and Rytova-Keldysh potential as a function of the electron-hole distance in a 2D

plane (dielectric constant = 4.5). Substantial difference is observed only for distances smaller than the 1s exciton Bohr radius ~ 2 nm. **b**, The exciton binding energy as a function of the principle quantum number for the Coulomb and the Rytova-Keldysh potentials. Small difference is observed for $n \geq 2$.

2. How is the spectral weight integration window of the 2s peak defined? Is any fitting procedure involved? Since the dash lines in Fig. 3c does not align with the baseline of the spectra, the authors may want to provide a slightly more detailed explanation in the method.

Response 10:

We thank the reviewer for the question. We chose an integration window of 5 nm, which covers the main reflectance peak of the 2s exciton. We have not performed any peak fitting because of the additional satellite features in addition to the main peak. Instead, we defined a straight line that connects the end points of the integration window as a baseline, and integrated the area above it (e.g. Fig. 3c in the main text). To verify the reliability of the procedure, we have varied the size of the spectral window (Fig. R2). The electric field dependence of the normalized spectral weight is independent of the size of the spectral window. We have included additional details on the spectral weight analysis in the revised Methods (page 6).

Figure R2. Electric field dependence of the normalized spectral weight at filling factor $\nu = 3$ and 3.6 K. The size of the spectral window varies between 4.5 and 5.5 nm.

REVIEWERS' COMMENTS

Reviewer #1 (Remarks to the Author):

I thank the authors for their careful response to all my questions. All my outstanding concerns have been answered satisfactorily, and therefore I suggest prompt publication of the work.

Reviewer #2 (Remarks to the Author):

I would like to recommend this paper to be accepted by Nature Communications for the following reasons.

All my technical comments in the last round have been properly addressed by the authors.

In terms of novelty, the authors of this work use an optical sensing technique to push the boundary of the field – they are now able to observe a continuous phase transition associated with the fractional insulating state, which was obscured by the contact resistance in their previous electrical experiment published in Nature. In response to my comment and the other referee's, the authors point out it is theoretically not clear how such a phase transition can happen – two potential physical mechanisms can possibly account for this continuous Mott-Wigner transition. Although the understanding of the exact mechanism has not been figured out, which leaves another important outstanding question for the field, I think the report of the existence of such a MIT is sufficient at this stage to be published in Nature Communications.

Reviewer #1 (Remarks to the Author):

I thank the authors for their careful response to all my questions. All my outstanding concerns have been answered satisfactorily, and therefore I suggest prompt publication of the work.

Response 1:

Again, we thank the reviewer for the comments that help improve our manuscript.

Reviewer #2 (Remarks to the Author):

I would like to recommend this paper to be accepted by Nature Communications for the following reasons.

All my technical comments in the last round have been properly addressed by the authors.

In terms of novelty, the authors of this work use an optical sensing technique to push the boundary of the field – they are now able to observe a continuous phase transition associated with the fractional insulating state, which was obscured by the contact resistance in their previous electrical experiment published in Nature. In response to my comment and the other referee' s, the authors point out it is theoretically not clear how such a phase transition can happen – two potential physical mechanisms can possibly account for this continuous Mott-Wigner transition. Although the understanding of the exact mechanism has not been figured out, which leaves another important outstanding question for the field, I think the report of the existence of such a MIT is sufficient at this stage to be published in Nature Communications.

Response2:

Again, we thank the reviewer for the comments that help improve our manuscript.